# Temporal Tissue Remodeling in Volumetric Muscle Injury with Endothelial Cell-Laden Patterned Nanofibrillar Constructs

**DOI:** 10.3390/bioengineering11121269

**Published:** 2024-12-14

**Authors:** Krista M. Habing, Cynthia A. Alcazar, Nathaniel Dobson, Yong How Tan, Ngan F. Huang, Karina H. Nakayama

**Affiliations:** 1Department of Biomedical Engineering, Oregon Health & Science University, Portland, OR 97239, USA; habing@ohsu.edu (K.M.H.); alcazarc@ohsu.edu (C.A.A.); treydobson60@gmail.com (N.D.); tanyo@ohsu.edu (Y.H.T.); 2Wallace H. Coulter Department of Biomedical Engineering, Georgia Institute of Technology, Atlanta, GA 30332, USA; 3Department of Cardiothoracic Surgery, Stanford University, Palo Alto, CA 94304, USA; ngantina@stanford.edu; 4Center for Tissue Regeneration, Veterans Affairs Palo Alto Health Care System, Palo Alto, CA 94303, USA; 5Department of Orthopaedics and Rehabilitation, Oregon Health & Science University, Portland, OR 97239, USA

**Keywords:** musculoskeletal trauma, volumetric muscle loss, regeneration, endothelial cell patterning, collagen scaffold, temporal remodeling

## Abstract

A primary challenge following severe musculoskeletal trauma is incomplete muscle regeneration. Current therapies often fail to heal damaged muscle due to dysregulated healing programs and insufficient revascularization early in the repair process. There is a limited understanding of the temporal changes that occur during the early stages of muscle remodeling in response to engineered therapies. Previous work demonstrated that nanotopographically patterned scaffolds provide cytoskeletal guidance and direct endothelial angiogenic and anti-inflammatory phenotypes. The aim of this study was to evaluate how endothelial cell (EC) patterning guides temporal and histomorphological muscle remodeling after muscle injury. In the current study, mice were treated with EC-laden engineered constructs that exhibited either aligned or random patterning of collagen nanofibrils, following a volumetric muscle loss injury (VML). Remodeling was evaluated at 2, 7, and 21 days post injury. Over the 21-day study, all groups (Acellular Aligned, EC Aligned, EC Random) demonstrated similar significant increases in vascular density and myogenesis. Animals treated with acellular controls demonstrated a two-fold decrease in muscle cross-sectional area between days 2 and 21 post injury, consistent with VML-induced muscle atrophy; however, animals treated with patterned EC-laden constructs exhibited preservation of muscle mass. The implantation of an EC-laden construct led to a 50% increase in the number of animals exhibiting areas of fibrous remodeling adjacent to the construct, along with greater collagen deposition (*p* < 0.01) compared to acellular controls 21 days post injury. These findings suggest that nanotopographically patterned EC-laden constructs may guide early muscle-protective programs that support muscle mass retention through myo-vascular independent pathways.

## 1. Introduction

Skeletal muscle repair is a tightly regulated process consisting of three distinct yet overlapping phases: (1) degeneration/inflammation, (2) regeneration/vascularization, and (3) remodeling [1,2]. The precise progression of these events is controlled by complex spatial and temporal cellular interactions within the skeletal muscle microenvironment. However, this repair process becomes dysregulated in cases of severe muscle injury such as volumetric muscle loss (VML), in which more than 20% of native muscle volume is lost to injury [3]. VML disrupts endogenous repair programs and cellular interactions, leading to a permanent loss of both muscle structure and function. Current clinical strategies for the surgical management of VML injuries, such as the transfer of free or adjacent muscle flaps, have been proven to save limbs yet fall short in restoring the skeletal muscle microenvironment or muscle function [4,5,6]. This results in approximately half of severe muscle trauma patients being left with chronic pain and permanent disability [7].

Regenerative therapeutics are poised to revolutionize the clinical management of VML injuries and highlight the importance of assisting native programs in restoring key cellular interactions to facilitate muscle healing. One critical cellular interaction is between skeletal muscle and vascular endothelial cells (ECs). ECs have been extensively utilized in the engineering of new vascular networks to restore blood flow to injured tissues and have been paired with material therapeutics to enhance recovery from VML [8,9,10]. The restoration of a vascular supply is a key first step in tissue healing, as it ensures the delivery of oxygen and nutrients essential for maintaining the viability and regenerative capacity of both native musculature and implanted therapeutics. The highly metabolic nature of skeletal muscle, particularly during periods of increased activity or repair, necessitates a dense and organized vascular network. Increasing vascular metrics, such as capillary density and the capillary-to-fiber ratio, are positively correlated with muscle function [11]. Furthermore, in the absence of sufficient vascular elements, cell-laden constructs are prone to necrotic core formation due to the limits of oxygen diffusion [12].

ECs also play a supportive role in muscle healing through paracrine guidance and the release of myogenic and angiogenic cytokines [13,14,15,16,17]. In vitro, ECs have been shown to improve myogenic cell viability, differentiation, and contractile function. In vivo, ECs have enhanced the engraftment of a muscle cell-laden engineered construct within a muscle defect and promoted the recruitment of host vasculature without directly contributing to the vasculature itself [18,19]. The production of paracrine factors can be further manipulated through substrate patterning. Environmental cues from underlying materials can modulate EC phenotypes and behavior [20,21,22]. Previous work has demonstrated that aligned patterning of nanofibrillar substrates increases the release of pro-myogenic and angiogenic cytokines when compared to random pattering [19,23,24]. This further supports the paradigm that ECs can act through the paracrine guidance of muscle cells to facilitate the healing process. However, the direct and exclusive effects of EC patterning in engineered therapeutics on endogenous muscle repair has not been fully established.

Paracrine signaling between the muscle and support cells is tightly regulated and involves the continuous modulation of cytokines throughout each phase of the repair process. It remains unclear which phase(s) of the repair process are most effectively supported by EC-laden constructs. Therefore, the aim of the current study was to histologically (1) determine if EC-laden constructs guide endogenous myogenic cell remodeling; (2) assess if endothelial patterning influences muscle remodeling; and (3) characterize the temporal changes and tissue response to injury following treatment with an engineered construct.

## 2. Materials and Methods

### 2.1. Experimental Design

The objectives of this study were to evaluate the role of patterned endothelial (EC)-laden engineered constructs in promoting muscle remodeling in a murine model of volumetric muscle loss (VML) and to characterize the temporal changes that occur during remodeling in response to these engineered therapies. Constructs were fabricated using aligned or random collagen nanofibrils and characterized by scanning electron microscopy. Three treatments were evaluated: (1) Acellular Aligned, (2) EC Random, and (3) EC Aligned. EC-laden constructs were cellularized with primary mouse dermal microvascular endothelial cells (DMECs). Muscle remodeling was assessed at three independent timepoints—2, 7, and 21 days post injury—using separate cohorts of animals for each timepoint. At all timepoints, muscle size and immunohistological markers for myogenesis and vascularization were analyzed. Fibrosis and fibrous encapsulation were evaluated only at the later timepoints of 7 and 21 days post injury (Figure 1a–d).

### 2.2. Fabrication of Nanofibrillar Collagen Constructs with Aligned and Random Fibril Patterning

The fabrication of the nanofibrillar collagen constructs has been described previously [19,24,25,26]. Briefly, 30 mg/mL collagen was obtained by dialyzing 10 mg/mL rat-tail collagen type I (Corning, Corning, NY, USA, 354249) for 35 min at 4 °C in Seamless Cellulose Dialysis Tubing (Fisher Scientific, Waltham, MA, USA, S25645G) on a bed of polyethylene glycol flakes (Fisher Scientific, BP233-1). Constructs with aligned fibril patterning were fabricated under shear by extruding high-density collagen from a blunt 22 G needle at an approximately 30° angle and a velocity of 340 mm/s into warmed 10X phosphate-buffered saline (PBS, pH 7.4, 37 °C) to instantaneously initiate fibrillogenesis as the collagen was extruded. Eight extruded aligned collagen strips (1 mm diameters) were assembled in parallel on a glass chip to create the collagen base of the transplantable construct (1 mm × 8 mm × 22 mm). Similar-sized constructs with random fibril patterning were fabricated without shear, which was achieved by extruding collagen under dry conditions followed by subsequent submersion in 10X PBS (Figure 1a). The constructs were partially dried for 2 h, washed in 1X PBS, and then fully dried overnight in a laminar flow hood.

### 2.3. Scanning Electron Microscopy of Nanofibrillar Collagen

The collagen constructs were prepared for scanning electron microscopy (SEM) through mounting on Indium Tin Oxide (ITO, SPI Supplies, West Chester, PA, USA, 06462-AB) coverslips. Initially, the constructs underwent fixation in 2.5% glutaraldehyde in 0.5 M sodium cacodylate buffer (Electron Microscopy Sciences, Hatfield, PA, USA, 50-980-232). The constructs were postfixed with 2% aqueous osmium tetroxide (Ted Pella, Redding, CA, USA, 18463), with washing in between using deionized water. Following fixation, the constructs were dehydrated first using successive ethanol washes and then by placement in a Leica CPD300 critical-point dryer (Leica Microsystems, Deerfield, IL, USA). The dehydrated constructs were then mounted on aluminum pins with carbon tabs and coated with an 8 nm thick layer of carbon in a Leica ACE600 sputter coater (Leica Microsystems). Additional silver paint was applied around the ITO coverslips to maximize conductivity around the edges of the construct. Imaging was conducted using a FEI Helios G3 NanoLab DualBeam electron microscope. The beam conditions were set to 2 keV, 25–100 pA, and a 4–5 mm working distance. The images were captured at 50,000X magnification and saved using a 6144 × 4096 image size and 1 µs pixel dwell time. ImageJ’s (Version 1.54i) FibrilTool [27] and jPOR [28] plugins were used to quantify construct anisotropy and porosity, respectively. The fibril diameter was quantified manually with ImageJ’s line tool (N = 4).

### 2.4. Culture of Primary Mouse Dermal Microvascular Endothelial Cell Constructs

B57BL/6 primary mouse dermal microvascular endothelial cells (ECs, Cell Biologics, Chicago, IL, USA, C57-6064) were cultured according to the manufacturer’s recommendations. Briefly, cells were cultured in T75 flasks coated with Gelatin-Based Coating Solution (Cell Biologics, 6950) in EC Growth Medium (Complete Mouse Endothelial Cell Medium with Kit, Cell Biologics, M1168). The cells were grown on chamber slides (Nunc™ Lab-Tek™ Chamber Slide System, Fisher Scientific, 177380) until 80% confluency, after which the cells were fixed in 4% cold paraformaldehyde (PFA) for 15 min, followed by 2 washes with 1X PBS. Cells in chamber slides were stored at 4 °C until stained.

To prepare constructs for transplantation into mice, scaffolds were sterilized in 70% ethanol for 15 min, allowed to air dry for 10 min in a biosafety cabinet, followed by two washes in sterile 1X PBS. Roughly 500,000 ECs (P4-6) were seeded per scaffold and allowed to adhere for 48 h followed by transplantation.

### 2.5. Immunocytochemical Characterization of Dermal Microvascular Endothelial Cells

To characterize the cultured ECs, cells in chamber slides were immunocytochemically stained for endothelial marker CD31. Briefly, samples were stained with rat anti-mouse CD31 (BD Bioscience, Franklin Lakes, NJ, USA, BDB550274), followed by goat anti-rat Alexa Fluor 594 (Invitrogen, Waltham, MA, USA, A-11007), and then counterstained with Hoechst 33342 (BD Biosciences, BDB561908) to visualize nuclei. Cells were imaged at 20X in a hydrated state in PBS using a Zeiss LSM 770 confocal microscope (Figure 1b).

### 2.6. Transplantation of Constructs into a Mouse Volumetric Muscle Loss Injury Model

All animal procedures were approved by the Institutional Animal Care and Use Committee of Stanford University (Protocol Number: 32865) and Oregon Health and Science University (Protocol Number: TR01_IP00002839). C57BL/6J mice (male, 8–10 weeks, Jackson Laboratories, Strain: #000664, N = 36 legs) were anesthetized using 3% isoflurane and were administered pre-surgical subcutaneous analgesic (Buprenorphine Extended Release, 1 mg/kg) and antibiotic (Baytril, 5 mg/kg). Hair was removed from the hindlimbs using a depilatory cream, and the surgical area was wiped with iodine and ethanol to sterilize. Surgical volumetric muscle loss (VML) was induced as previously described [19,25,26]. In short, a 20–30% muscle defect was created bilaterally in both the right and left tibialis anterior (TA) muscles with a surgical ablation measuring approximately 7 mm × 2 mm × 2 mm at the TA midline. The following constructs were assessed: (1) Acellular Aligned, (2) EC Random, and (3) EC Aligned (N = 12 legs per group). Constructs were implanted in the bilateral defect sites and secured with 10-0 non-absorbable nylon suture loops at the distal and proximal ends of the defect. Muscle flaps were secured over the construct using an X suture with 8-0 non-absorbable nylons sutures followed by skin flap closure with a continuous suture of the same material (Figure 1c). Tissues were collected on days 2, 7, and 21 post injury to histologically assess remodeling (N = 4 legs per group per timepoint, Figure 1d).

### 2.7. Histological Analysis of Blood Perfusion

On days 2, 7, and 21 post VML injury, mice were injected via the tail vein with 200 µL of isolectin (GS-IB4 from Griffonia Simpicifolia, Alexa Fluor 647 conjugate, 100 µg/mL, Invitrogen, I32450), a fluorescently labeled endothelial binding protein that was used to identify perfused blood vessels. Following the injection, mice were euthanized using 1% carbon dioxide and cervical dislocation. The TA muscles were excised and fixed on a rocker in 0.2% PFA at 4 °C overnight. Following initial fixation, muscles were saturated with 20% sucrose to achieve density equilibrium and then embedded for cryosectioning in optical cutting temperature compound and snap frozen. For immunofluorescent imaging, 10 µm thick transverse tissue sections underwent further fixation in 4% PFA, followed by mounting with Vectashield plus DAPI (Vector Laboratories, Newark, CA, USA, H-1200-10). All images were taken using a Zeiss LSM 900 confocal microscope, and tiled 20X images were used to capture the entire tissue cross-section.

Blood vessels were quantified semi-automatically using ImageJ and Cellpose 2.0 [29]. In ImageJ, a region of interest (ROI) was created that extended 500 µm from the border of the implanted construct. The isolectin channel was false-colored red and then inserted into Cellpose, and a custom model was run to segment each vessel. The custom model was created from Cellpose’s pre-trained “Nuclei” model that was then fine-tuned using user input on four isolectin training images. Blood vessel quantification was expressed as the total number of isolectin-positive cells per square millimeter or the average number of isolection-positive cells per myofiber (N = 4 legs per group per timepoint).

### 2.8. Histological Analysis of Myogenesis

The quantification of myofiber regeneration was performed using immunofluorescent staining with rabbit anti-laminin (1:100, overnight, 4C, Abcam, Waltham, MA, USA, ab11575), a basement membrane protein, that was then conjugated to goat anti-rabbit Alexa Fluor 594 (1:200, 1 h, room temperature). Tissues were mounted using VectaShield plus DAPI. All images were taken using a Zeiss LSM 900 confocal microscope, and tiled 20X images were used to capture the entire tissue cross-section. Newly regenerated de novo myofibers were identified as cells with borders positive for laminin staining and with centrally located nuclei. To effectively capture regeneration, a region of interest was marked for each tissue section that extended 500 µm from the border of the implanted construct. The de novo myofiber density was expressed as the number of myofibers with centrally located nuclei per square millimeter and as a percentage of total myofibers.

Within the region of interest, the individual area of all myofibers was quantified using Cellpose 2.0 [29]. The laminin channel was inserted into Cellpose, and a custom model was run to segment each myofiber. The custom model was created using Cellpose’s pre-trained “Cyto” model that was then fine-tuned using user input on four laminin training images. Masks of segmented myofibers were saved and then overlayed on the original image as the ROI’s in ImageJ using the Labels to ROIs plugin [30]. The area of each segmented fiber was measured and then averaged for each mouse. Myofiber masks were pseudo-colored using the MorphoLibJ plugin to facilitate easier visualization of individual myofibers (N = 4 legs per group per timepoint).

### 2.9. Histological Assessment of Muscle Histomorphology

Tissue sections were stained with routine hematoxylin and eosin (N = 4 legs per group per timepoint) or Masson’s Goldner Trichrome staining (N = 4 legs per group, day 21 only) to evaluate tissue morphology and fibrosis, respectively. Brightfield images were taken by Oregon Health & Science University’s Advanced Light Microscopy Core at 20X using a Zeiss Axioscan Slide Scanner. In the H&E images, the TA cross-sectional area was measured using the ImageJ polygon tool, and construct fibrous encapsulation was determined through visual assessment. Cross-sectional area measurements were consistently obtained from sections located near the center of the tissue. To ensure repeatability, the center of the tissue was taken as 2.5 mm from the proximal end of the tissue. Collagen deposition was quantified using a combination of custom MATLAB code (Version R2023b) and the Color Deconvolution tool in ImageJ. Firstly, Masson’s Goldner images were preprocessed using the MATLAB code to subtract image background. The cleaned images were then opened in ImageJ and underwent Masson’s Trichrome Color Deconvolution, a technique for unmixing brightfield images into channels that represent the absorbance of specific dyes. Collagen deposition was isolated by adjusting the threshold of the deconvolved red channel. The area of collagen deposition was then expressed as a percentage of the total TA muscle area.

### 2.10. Statistical Analysis and Rigor

All statistical analysis was completed using GraphPad Prism (Version 10.4.1). Ordinary two-way ANOVAs with post hoc Tukey’s adjustment using a single pooled variance were used when comparing timepoints and experimental groups. For collagen deposition, which was solely investigated at the final timepoint, a one-way ANOVA with post hoc Tukey’s adjustment was used to compare the groups. All groups and timepoints display N = 4, indicating the number of legs quantified. A total of 18 animals were used, with 2 animals per group per timepoint, as the VML injury was bilateral, resulting in 4 legs per group per timepoint. Significance was taken at *p* < 0.05 (*), *p* < 0.01 (**), *p* < 0.001 (***), and *p* < 0.0001 (****). All graphs display the mean ± standard deviation (SD). No animals or legs were excluded from analysis. The surgeon and lead investigator were unblinded for the sake of analysis. However, the individuals who performed much of the histological quantification were blinded to prevent bias during the selection of positively stained observations. All animals were randomly assigned to one of the three treatment groups.

## 3. Results

### 3.1. Characterization of the Engineered Construct

Aligned or randomly patterned nanofibrillar collagen constructs were fabricated using a shear-based extrusion method (Figure 1a). The construct anisotropy, porosity, and fibril diameter were characterized using scanning electron microscopy (SEM) (Figure 2a). The aligned and randomly patterned constructs exhibited overall anisotropy values of 0.189 ± 0.097 and 0.752 ± 0.061, respectively (*p* < 0.0001, Figure 2b). Anisotropy values closer to one denote alignment lengthwise along the construct. Both construct patterns demonstrated similar levels of porosity, though the porosity of the randomly patterned construct (11.2 ± 2.81%) was slightly higher than that of the aligned construct (8.53 ± 0.979%, Figure 2c). The random constructs exhibited 1.45-fold thicker nanofibrils (72.6 ± 6.11 nm) than the aligned constructs (50.0 ± 7.86 nm, *p* = 0.004, Figure 2d). Constructs were seeded with primary mouse dermal microvascular endothelial cells (ECs). The cell population was confirmed using immunofluorescent staining with CD31, a cell surface marker for endothelial cells (Figure 1b).

### 3.2. Development of VML Model

To evaluate how endothelial transplantation and construct patterning influences muscle remodeling following severe muscle trauma and to outline a timeline for healing events, a VML injury was surgically created in the tibialis anterior muscle (TA, Figure 1c). Mice were divided into cohorts that received one of the following constructs: (1) Acellular Aligned, (2) EC Random, or (3) EC Aligned. Tissues were collected on days 2, 7, and 21 post injury to histologically assess remodeling (Figure 1d).

### 3.3. Muscle Vascularization

Revascularization is a critical step in supporting proper muscle remodeling following VML injury. This was assessed histologically 2, 7, and 21 days post injury by identifying the density of blood vessels that were positive for isolectin, an endothelial binding protein, within the defect site (500 µm radius around the construct region) (Figure 3a). By day 21 post injury, all groups saw significant increases in vascular density compared to day 2 (*p* < 0.0001–0.001) and day 7 (*p* < 0.001–0.05). On day 7 post injury, the EC Random constructs demonstrated early advances in vascular density (782 ± 202 blood vessels/mm^2^) compared to day 2 (345 ± 183 blood vessels/mm^2^, *p* = 0.055). However, by day 21, both the Acellular Aligned (1720 ± 280 blood vessels/mm^2^) and EC Aligned constructs (1748 ± 459 blood vessels/mm^2^) exhibited significantly greater vascular densities than the EC Random constructs (1231 ± 110 blood vessels/mm^2^), with a 1.4-fold greater vessel density (*p* < 0.05, Figure 3b).

Although blood vessel density increased significantly across all groups over 21 days, a significant increase in the vessel-to-myofiber ratio was observed only in the Acellular Aligned group between days 2 (1.29 ± 0.81 vessels/myofiber) and 21 (2.25 ± 0.50 vessels/myofiber, *p* = 0.045) as well as between days 7 (1.17 ± 0.50 vessels/myofiber) and 21 (*p* = 0.022). The vessel-to-myofiber ratio exhibited similar trends to those of vascular density in the EC-laden groups. By day 7 post injury, the EC Random constructs exhibited a modest, early, 1.3-fold increase in the vessel-to-myofiber ratio (1.27 ± 0.303 vessels/myofiber) compared to day 2 post injury (0.947 ± 0.648 vessels/myofiber). The EC Random group vessel-to-myofiber ratio continued to increase through day 21 post injury (1.88 ± 0.517 vessels/myofiber) but was surpassed by the vessel-to-myofiber ratio of the Acellular Aligned and EC Aligned groups (2.28 ± 0.558 vessels/myofiber, Figure 3c). Taken together, this temporal analysis of revascularization suggests that while EC Random constructs exhibit early advantages, aligned constructs, irrespective of cellularization, better support sustained revascularization following VML injuries.

### 3.4. Muscle Myogenesis

To determine how endothelial orientation influences myogenesis, the density of de novo myofibers, individual myofiber cross-sectional area (CSA), and TA muscle CSA were quantified. Immunofluorescent laminin staining was used to demarcate the borders of individual myofibers, and myonuclear relocation (centrally located nuclei) served as an indicator for actively regenerating de novo myofibers (Figure 4a, left). Within the defect site (500 µm radius around the construct region), all groups demonstrated similar percentages of de novo myofibers across all timepoints. The greatest increase in the percentage of de novo myofibers occurred between days 2 (~12% de novo myofibers) and 7 post injury (~50% de novo myofibers) with a 3.7- to 4.7-fold increase (*p* < 0.0001–0.001) depending on treatment group. Following day 7, the percentage of de novo myofibers was maintained at 40–68% through the study endpoint, day 21 post injury (Figure 4b).

Despite considerable variance, the pronounced increase in de novo myofibers between days 2 (~46 myofibers/mm^2^) and 7 post injury (~334 myofibers/mm^2^) was also apparent when examined as de novo myofiber density per unit area, demonstrating a 6.6- to 8.1-fold increase depending on treatment group. Unlike the examination of the percentage of de novo myofibers, the density of de novo myofibers in the EC-laden aligned group continued to increase between days 7 (347 ± 142 myofibers/mm^2^) and 21 (941 ± 434 myofibers/mm^2^, *p* = 0.0007). On day 21, the de novo myofiber density of the EC Aligned group surpassed that of the Acellular Aligned (529 ± 122 myofibers/mm^2^, *p* = 0.0178) and EC Random groups (409 ± 189 myofibers/mm^2^, *p* = 0.0021, Figure 4c). This discrepancy between the percentage and density of de novo myofibers on day 21 is likely in part due to changes in myofiber CSA.

Myofiber CSA, a secondary indicator of myogenesis, was assessed using Cellpose, a cellular segmentation algorithm (Figure 4a, right) [29]. Both the average myofiber CSA and the frequency distribution of myofiber CSAs displayed marked reductions in overall myofiber size over time across all treatment groups, consistent with a higher percentage of newly formed and actively regenerating myofibers. On day 2 post injury, the average myofiber CSA of the EC-laden constructs, regardless of patterning, was significantly larger than that of the acellular constructs (*p* < 0.01–0.05). A significant decline in myofiber CSA was observed in all groups between days 2 and 7 post injury with the EC-laden groups both displaying more than a two-fold decrease (*p* < 0.0001–0.001). Between days 7 and 21 post injury, myofiber CSAs exhibited a relatively modest additional decrease across all treatment groups, ranging from 1.1- to 1.4-fold (Figure 4d). The frequency distribution of myofiber CSAs additionally demonstrates the marked shift toward smaller myofibers over the study’s time course (Figure 4e). These findings suggest that myogenesis initiates within the first 7 days post injury, yet the process is not fully resolved by day 21, as evidenced by the continued presence of small myofibers displaying signs of ongoing nuclear relocation. Moreover, these findings suggest that the rate and extent of myogenesis occurs largely independently of treatment.

H&E-stained tibialis anterior (TA) muscle cross-sections revealed that, despite similarities in myogenesis on the cellular scale across treatment groups, the overall TA muscle CSA was influenced by the treatment group over the study time course (Figure 5a). From days 2 to 7 post injury, the TA muscle CSA remained consistent across both aligned treatment groups. However, treatment with an EC Random construct resulted in a 1.5-fold increase in area, increasing from 5.43 ± 0.72 mm^2^ to 8.22 ± 1.37 mm^2^ (*p* = 0.0149). Subsequently, from day 7 to 21 post injury, EC-laden constructs, regardless of patterning, maintained their CSAs, while treatment with the Acellular Aligned construct saw a two-fold decrease in TA muscle CSA from 6.72 ± 1.48 mm^2^ to 3.32 ± 0.94 mm^2^ (*p* = 0.0029), indicative of VML-induced muscle atrophy. On day 21 post injury, because of this atrophy, tissues treated with an Acellular Aligned construct had significantly smaller TA muscle CSAs compared to those treated with an EC Aligned construct (6.02 ± 1.33 mm^2^, *p* = 0.0185) or an EC Random construct (6.19 ± 1.80 mm^2^, *p* = 0.0118, Figure 5b). While minimal differences were observed in myogenesis at the molecular level, these results emphasize the role of construct endothelialization in promoting comprehensive preservation of muscle mass following VML.

### 3.5. Muscle Collagen Content and Construct Fibrous Encapsulation

Investigating fibrosis is crucial when utilizing cell-laden biomaterials, such as EC-laden constructs, as the introduction of exogenous cells can trigger an inflammatory foreign body response. Such a response leads to the formation of fibrovascular tissue on and around the transplanted material. Collagen deposition, and consequently fibrosis, was investigated using Masson’s Goldner staining (Figure 6a). At the overall scale of the TA muscle, the cellularization of the construct, irrespective of its patterning, resulted in upwards of a two-fold increase in the percentage of collagen deposition within the TA muscle at 21 days post injury. Specifically, the percentage of collagen increased from 7.44 ± 2.44% for the Acellular Aligned construct-treated tissues to 14.9 ± 1.52% and 13.4 ± 2.21% for the EC Random and Aligned construct-treated tissues, respectively (*p* < 0.01, Figure 6b). In addition to exhibiting greater levels of collagen deposition, all tissues that received an EC-laden construct at both 7 and 21 days post injury displayed fibrous encapsulation of the construct. For the tissues with Acellular Aligned constructs, encapsulation was observed in 75% and 25% of tissues on days 7 and 21, respectively (Figure 6c,d). These results indicate that the implantation of EC-laden constructs, regardless of patterning, exerts a sizable and lasting effect on the fibrotic response of the tissue, which does not resolve by 21 days post injury, as observed with acellular constructs.

## 4. Discussion

Following VML injuries, normal physiological muscle repair is impaired due to the loss of the spatial and temporal cellular cues, which are crucial for driving the repair cascade. Regenerative treatments aim to restore these cues utilizing constructs that provide both spatial and cellular guidance. Previous work has demonstrated that, following a VML injury, construct efficacy was enhanced through the co-culture of myogenic cells with ECs. The ECs played a supportive paracrine role, and their effectiveness was contingent upon nanotopographical patterning [19]. Given that ECs play a complimentary role within myogenic constructs, the current study aimed to assess if patterned ECs alone may augment endogenous skeletal muscle remodeling following a bilateral VML injury while also working to establish a timeline for the repair response when an engineered construct is utilized.

Despite differences in construct patterning and endothelialization, immunohistological analysis demonstrated no significant variations in the early stages of endogenous muscle repair among the Acellular Aligned, EC Random, and EC Aligned construct groups. Acellular Random constructs were not included in this study, as previous findings indicated that the patterning of acellular constructs has minimal impact on myo-vascular outcomes [19,25]. Including this condition could have reinforced existing findings and provided greater context. However, the study design was streamlined to more specifically focus on investigating myo-vascular outcomes associated with EC-laden patterned constructs. Future research could incorporate this group to facilitate more direct comparisons. By day 21, the EC Random group exhibited lower vascular density compared to the Acellular Aligned and EC Aligned groups. Interestingly, myogenesis remained consistent across all groups. Given the established feedback between vascularization and myogenesis [17], it might be expected that lower vascular density would inhibit myogenesis. However, early revascularization levels may have been sufficient to support myogenesis, or this suggests that additional factors beyond vascularization, such as the immune response, play a key role in determining repair outcomes following VML injuries.

These myo-vascular findings contrast with prior work where differences in both construct patterning and cellularization significantly influenced angiogenesis and myogenesis [19]. In the present study, the observed similarity in outcomes could be attributed to several different mechanisms: (1) endothelial cells alone lack sufficient feedback to drive repair and (2) a foreign body response restricts the efficacy of the EC-laden constructs.

Muscle repair is driven, in part, by the intertwined processes of revascularization and myogenesis, which stimulate one another through the reciprocal interactions of ECs and myogenic cells [13,17,31]. This feedback system has previously been leveraged to improve the efficacy of engineered muscle constructs composed of a combination of ECs and myogenic cells. Within such constructs, ECs support sustained myogenic cell viability and expansion, subsequently enabling the constructs to more effectively improve the repair of the surrounding native muscle [18,19]. In the current study, however, the interaction between ECs and myogenic cells is more constrained. The transplanted ECs were restricted to interacting with native myogenic cells, as opposed to myogenic cells within the construct. Given the substantial size of VML injuries and the already limited population of satellite cells [32], the myogenic cells responsible for repair, transplanted ECs, likely have reduced capacity to directly influence muscle regeneration. As the EC-laden constructs did not substantially improve myogenesis compared to acellular constructs, this suggests that ECs may function most effectively as supportive, rather than primary components within engineered constructs.

It is crucial to consider the host foreign body response when designing cell-laden constructs as exogenous cells, and materials can trigger a detrimental immune reaction, thus impairing treatment efficacy. While this study did not directly investigate immune cells, fibrotic deposition indirectly reflects the foreign body response. Following construct transplantation, the immune system modulates fibrosis to protect the host tissue through construct encapsulation [33,34]. In the current study, EC-laden constructs were persistently encapsulated beginning 7 days post injury. Given that these ECs are likely functioning in a paracrine manner, isolating the constructs from the surrounding muscle may disrupt signaling, limiting therapeutic effects. This limitation may explain why the vascular density at day 21 in the EC Aligned group was comparable to that of the Acellular Aligned group, despite aligned ECs being known to secrete pro-angiogenic factors [19]. The EC Random group, in addition to being hindered by the foreign body response, may also be secreting fewer angiogenic cytokines and more pro-inflammatory cytokines due to the random orientation of the ECs. This imbalance in cytokine secretion could be exacerbating the immune response, further inhibiting angiogenesis and contributing to the lower vascular density observed in this group.

In previous work, construct encapsulation was not observed, likely due to differences in immune–construct crosstalk, as NOD-SCID mice and different exogenous cells were used [19]. Other exogeneous cell types may be less immunogenic than ECs, as EC surface proteins have been associated with allograft rejection [35,36,37]. Besides disrupting host–construct signaling, the foreign body response likely leads to the rapid destruction of transplanted ECs due to the activation of host phagocytes [33]. Other types of EC-laden treatments indicate that transplanted ECs only persist for 3–7 days post transplant in competent immune systems [38,39]. Thus, if EC-laden constructs are not only isolated from the muscle, but their cellularization only persists for a short period, the construct likely has a limited effect on the muscle repair response.

Beyond construct encapsulation, the transplantation of EC-laden constructs also led to increased fibrotic collagen deposition throughout TA muscle. VML injuries are a unique case, with growing evidence suggesting that post-injury fibrosis can be beneficial, termed functional fibrosis. This fibrosis not only serves as a bridge for force transmission between the uninjured areas of muscle, but also shields these areas from excessive mechanical stress, thus preventing further injury [40]. The functional implications of fibrosis in the present study remain unclear, and future research involving muscle physiology testing is needed. A limitation of this study is the reliance on fibrosis as a surrogate marker for the immune response. While it provides a broad overview of how the immune system responds to foreign materials, fibrosis does not capture how processes such as angiogenesis and myogenesis are intricately linked to immune activity, exerting significant influence over repair outcomes [41].

While the repair timeline for acute muscle injuries has been well characterized [42], timelines for VML repair, especially those looking at myo-vascular outcomes, remain limited. Establishing repair timelines for VML injuries is critical for understanding if severe injuries alter the repair dynamics that would further aid in therapeutic development. Regarding revascularization, the current study shows that significant increases in vascular density were not evident until days 7 to 21 post injury. This contrasts with another study that observed significant vascularization prior to this timepoint [43]. Interestingly, differing vascular markers may explain this variation. The current study used isolectin, marking only perfused vessels, while the contrasting study employed CD31, marking all vessels and endothelial sprouts. Vessels must develop prior to perfusion, likely explaining the slower timeline when isolectin is employed as opposed to CD31. The use of both markers in future work would allow for the development of a more comprehensive timeline of revascularization.

Temporal variations in myogenesis are also evident across VML models. In this study, de novo myofibers, demarcated by their centrally located nuclei, were prominent in the defect site beginning 2 to 7 days post injury, whereas in another model, myofiber infiltration does not begin until 14 days post injury [43]. Some differences that may contribute to these variations in myogenesis include (1) the use of partial- versus full-thickness VML injuries, (2) VML size, (3) treatment versus no-treatment conditions, and (4) use of unilateral versus bilateral injuries. The bilateral injury model employed in this study may be considered a limitation, as it complicates the analysis of the effects of both injury burden and treatment. Additionally, the limited availability of previous bilateral injury studies restricts opportunities for the comparison and contextualization of findings. However, bilateral VML injury models may more accurately represent the often polytraumatic nature of clinical extremity injuries, where multiple extremities are affected, thus enhancing the translational relevance of the study’s findings [44].

Returning to the temporality of myogenesis, in the present study, myogenesis persisted upwards of 21 days post injury with no decline in the percentage or density of de novo myofibers or an increase in the myofiber cross-sectional area. The prolonged presence of myofibers with centrally located nuclei for 1–6 months post injury is a common observation in rodent muscle injury models [45,46]. However, it remains unclear whether this extended presence of de novo myofibers is an indicator of incomplete myogenesis in which a study timeline fails to capture the remodeling and maturation phase of muscle repair [1,2]. However, in the present study, incomplete regeneration is further corroborated by vascular data, as vascular density was still increasing by day 21 post injury. Upon the completion of regeneration, vascular density is typically expected to stabilize, if not decrease [43]. Future studies should extend the post-injury observation period to ensure that the full extent of muscle repair is captured.

## 5. Conclusions

This study histologically evaluated the role of patterned EC-laden constructs in guiding endogenous muscle remodeling and characterized the temporal tissue response to injury following treatment. These findings indicate that EC-laden constructs, regardless of patterning, have limited impact on muscle repair. However, based on prior research, ECs may still function effectively as support cells within constructs rather than as primary therapeutic agents. These results also revealed distinct temporal changes in tissue remodeling, differing from other studies and suggest that a 21-day timeline may be insufficient for capturing the full extent of the muscle remodeling process. While treatment did not drastically influence outcomes, this study underscores the need for future comparative analyses of VML models and extended observation periods to better track therapeutic effects. These insights will contribute to the advancement of biomaterial-based interventions for muscle regeneration and the enhancement of our understanding of VML pathophysiology.

## Figures and Tables

**Figure 1 bioengineering-11-01269-f001:**
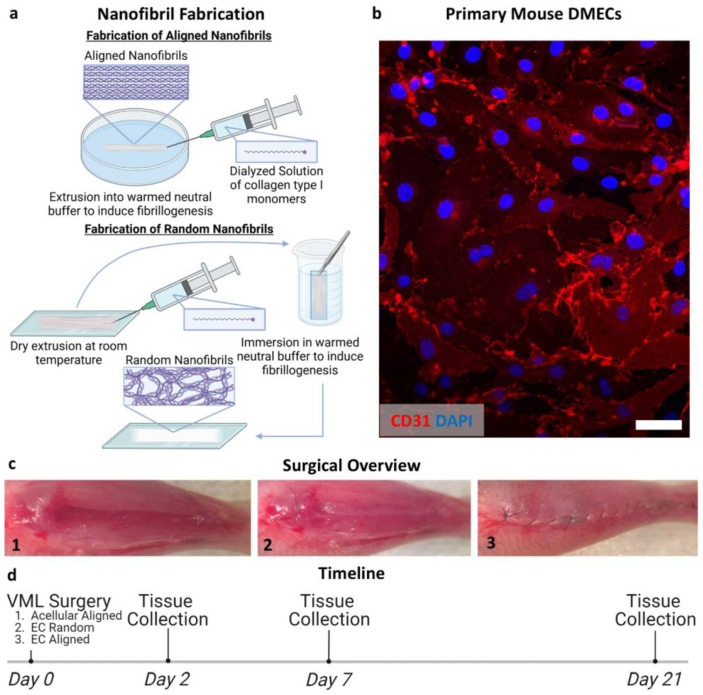
Overview of study design. (**a**) Diagram demonstrating aligned and random scaffold fabrication; (**b**) confocal microscopy image showing the primary mouse dermal microvascular endothelial cells (DMECs) stained with anti-CD31 (red) and DAPI. These were used to cellularize the engineered constructs. (**c**) VML surgical procedure depicting (1) 20–30% tibialis anterior muscle ablation, (2) implantation of an engineered construct, (3) muscle and skin closure over the implanted construct; (**d**) study timeline depicting division into 3 experimental groups based on treatment followed by tissue collection and immunohistochemistry on days 2, 7, and 21 post VML surgery. Scale bar: (**b**) 50 μm.

**Figure 2 bioengineering-11-01269-f002:**
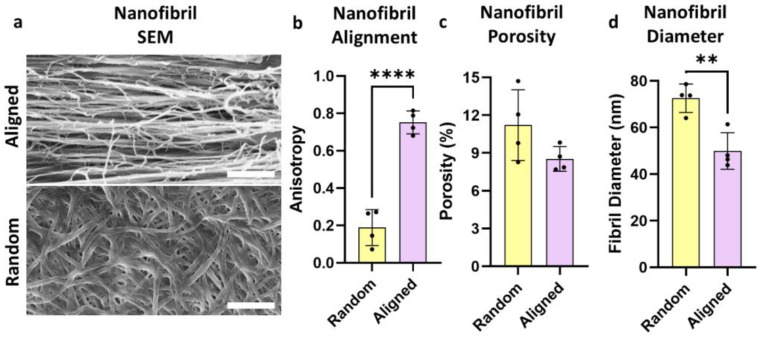
Scanning electron microscopy analysis of construct nanofibrillar properties. (**a**) Scanning electron microscopy images of the aligned (top) or random (bottom) collagen nanofibrils that make up the structural foundation of the constructs; (**b**–**d**) quantification of nanofibril anisotropy (**b**), porosity (**c**), and fibril diameter (**d**) for both construct patterns. Scale bars: (**a**) 1 μm. Significance was determined using an unpaired *t*-test with *p* < 0.01 (**) and *p* < 0.0001 (****). Values shown are mean ± SD.

**Figure 3 bioengineering-11-01269-f003:**
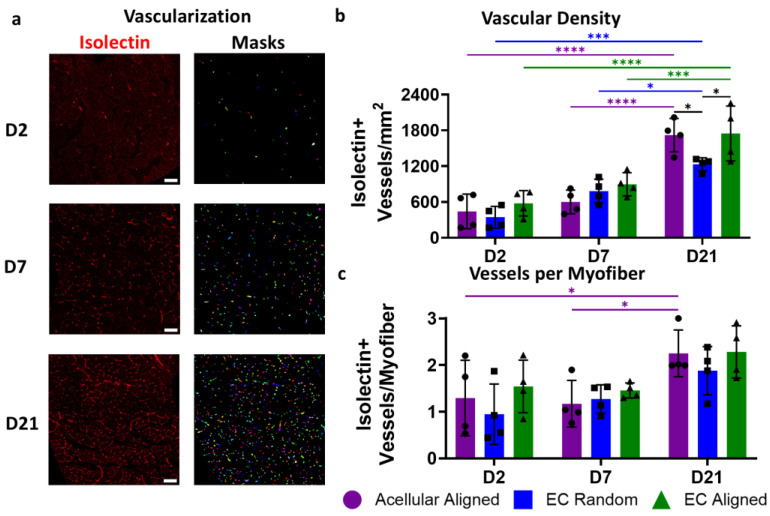
Timeline of vascularization following VML injury and treatment with patterned EC-laden constructs. (**a**) Representative confocal microscopy images of transverse cross-sections of TA muscle depicting isolectin-positive blood vessels (red) with mask overlays of individually segmented vessels generated by Cellpose on days 2, 7, and 21 post injury; (**b**,**c**) Quantification of vascular density (**b**) and vessel-to-myofiber ratio (**c**) within a 500 µm area from an Acellular Aligned, EC Random, or EC Aligned construct on days 2, 7, and 21 post injury (N = 4). Scale bars: (**a**) 50 µm. Significance was determined using a two-way ANOVA with *p* < 0.05 (*), *p* < 0.001 (***), and *p* < 0.0001 (****). Values shown are mean ± SD.

**Figure 4 bioengineering-11-01269-f004:**
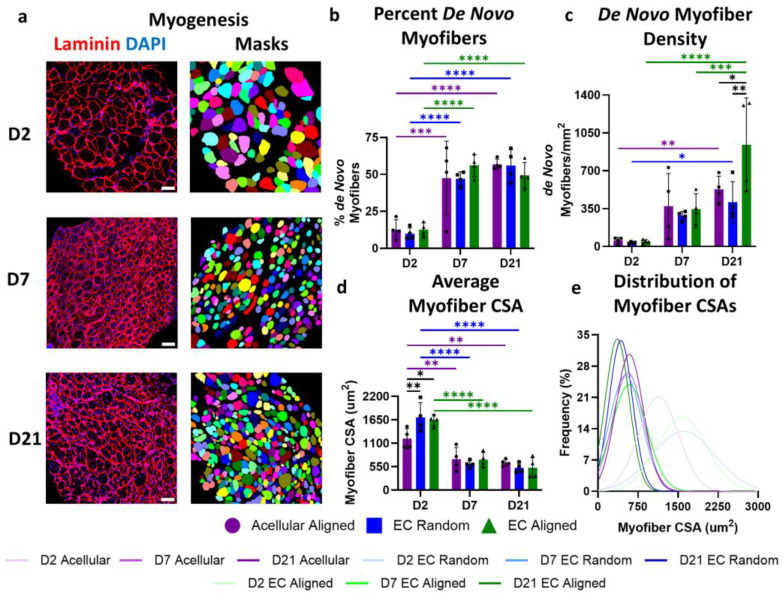
Timeline of myogenesis following VML injury and treatment with patterned EC-laden constructs. (**a**) Representative confocal microscopy images of transverse cross-sectionss of TA muscle depicting myofibers outlined by laminin (red) with mask overlays of individually segmental myofibers generated by Cellpose on days 2, 7, and 21 post injury. (**b**–**d**) Quantification of myogenesis based on the percent (**b**) and density (**c**) of de novo myofibers in addition to average myofiber cross-sectional area (CSA) (**d**) within a 500 µm area from an Acellular Aligned, EC Random, or EC Aligned construct on days 2, 7, and 21 post injury (N = 4). (**e**) Frequency distributions of myofiber CSAs within a 500 µm area from an Acellular Aligned, EC Random, or EC Aligned construct on days 2, 7, and 21 post injury. Scale bars: (**a**) 50 µm. Significance was determined using a two-way ANOVA with *p* < 0.05 (*), *p* < 0.01 (**), *p* < 0.001 (***), and *p* < 0.0001 (****). Values shown are mean ± SD.

**Figure 5 bioengineering-11-01269-f005:**
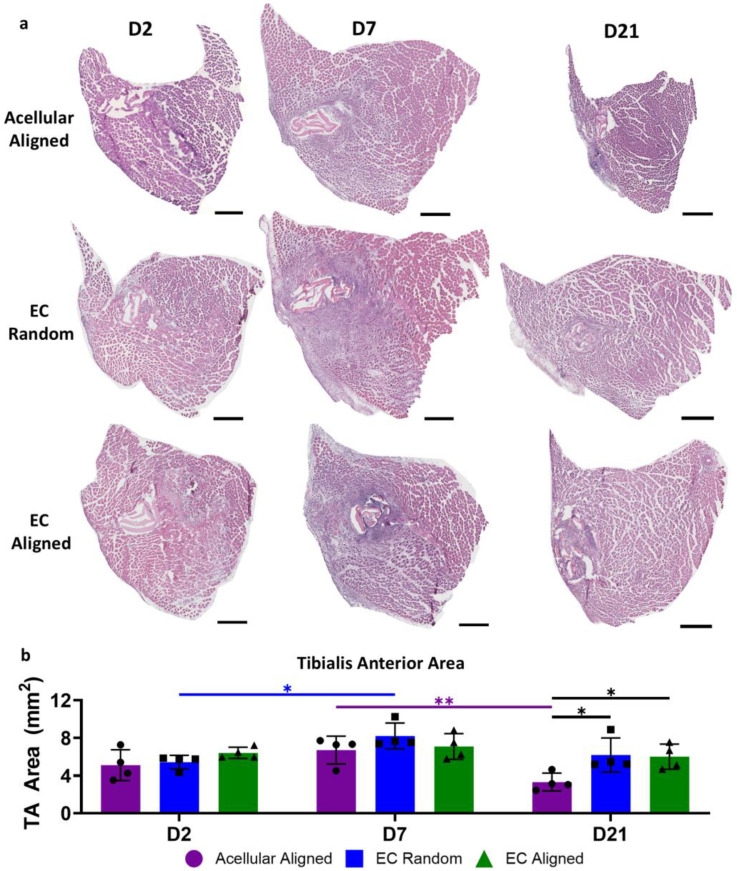
Timeline of TA muscle cross-sectional area following VML injury and treatment with patterned EC-laden constructs. (**a**) Representative hematoxylin and eosin (H&E) staining of transverse TA muscle sections collected on days 2, 7, and 21 post injury from mice, which received an Acellular Aligned, EC Random, or EC Aligned construct. (**b**) Quantification of average TA muscle cross-sectional areas on days 2, 7, and 21 post injury in mice, which received an Acellular Aligned, EC Random, or EC Aligned construct (N = 4). Scale bars: (**a**) 500 µm. Significance was determined using a two-way ANOVA with *p* < 0.05 (*) and *p* < 0.01 (**). Values shown are mean ± SD.

**Figure 6 bioengineering-11-01269-f006:**
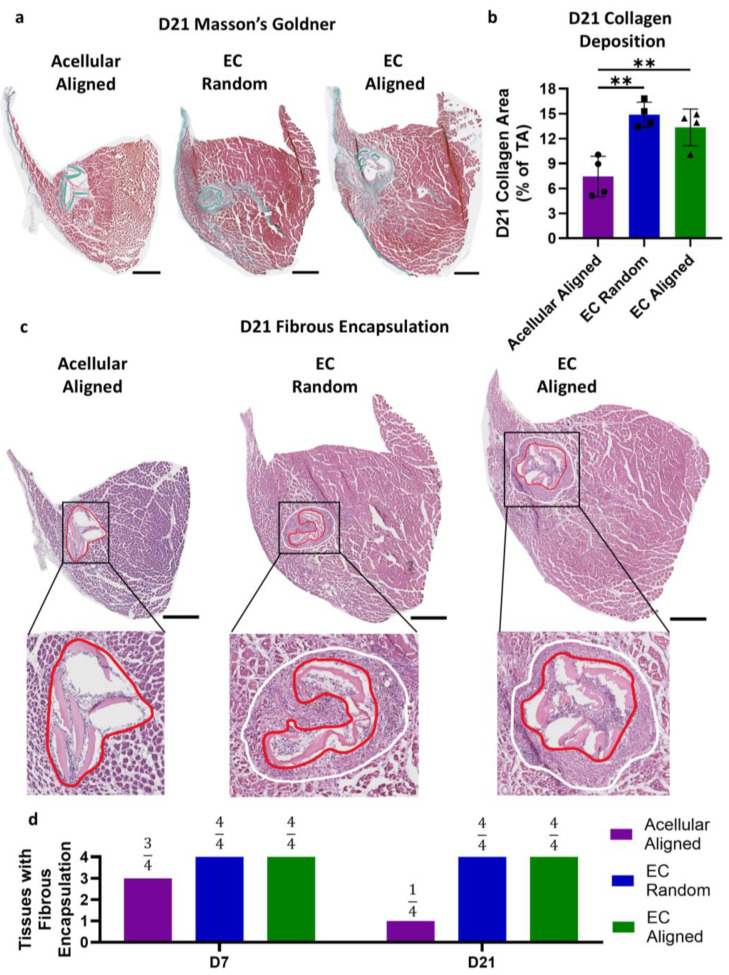
TA muscle fibrosis and construct encapsulation following VML injury and treatment with patterned EC-laden constructs. (**a**) Masson’s Goldner staining of representative TA muscle cross-sections with Acellular Aligned, EC Random, or EC Aligned constructs 21 days post injury; (**b**) percent area positive for collagen relative to TA size 21 days post injury in mice that received an Acellular Aligned, EC Random, or EC Aligned construct (N = 4); (**c**) representative H&E images of transverse TA muscle sections with Acellular Aligned, EC Random, or EC Aligned constructs depicting construct fibrous encapsulation, or lack thereof, 21 days post injury. The construct is outlined in red, and the fibrous encapsulation is outlined in white. (**d**) Quantification of the number of transverse TA muscle sections presenting with fibrous encapsulation on days 7 and 21 post injury in mice that received an Acellular Aligned, EC Random, or EC Aligned construct (N = 4). Scale bars: (**a**,**c**) 500 µm. Significance was determined using a one-way ANOVA with *p* < 0.01 (**). Values shown are mean ± SD (**b**) or tissue encapsulation count (**d**).

## Data Availability

The data that support the findings of this study will be available from the corresponding author upon reasonable request.

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
