# Peer review of "Temporal Tissue Remodeling in Volumetric Muscle Injury with Endothelial Cell-Laden Patterned Nanofibrillar Constructs"

_bioengineering, 2024, doi:10.3390/bioengineering11121269_

Round 1

Reviewer 1 Report

Comments and Suggestions for Authors

Specific comments: The article presents an interesting opportunity to expand information about the context. Nevertheless, it is imperative to take into account a number of variables:

Title: The title is clear and expresses conformity with the study proposal.

The Abstract is competently written and provides a lucid overview of the study's methodology and findings.

The Introduction is well-written and presents a clear definition of the problem. Nevertheless, a point is worthy of further consideration:

In what theoretical framework does the sentence in lines 58 to 60 find its justification?

Material and Methods

It is well written but it is essential to commence the Material and Methods section by delineating the type of study (experimental or otherwise), temporality (cross-sectional or otherwise), prospective or otherwise, and other pertinent characteristics. Additionally, it is imperative to furnish details regarding the ethical approval of the study, which can be found in section 2.5. entitled “Transplantation of Constructs into a Mouse Volumetric Muscle Loss Injury Mode”. It is recommended that this information be included at the beginning of the Materials and Methods section.

It would be important to have a section for describing the sample/animals. Information on this appears in the ‘Statistical Analysis and Rigour’ section on lines 220-223 (All groups and timepoints display N = 4, … resulting in 4 legs per group per timepoint). In this context, please clarify whether the sample size or power was calculated. Besides, it would be beneficial to include further details regarding the sample, such as the age of the animals, sex and other characteristics.

It would be important for there to be a figure showing the design of the study, something similar to Figure 1 presented, but not with results.

Results:

In general Results session it is somewhat confusing because much of what is described as results should be described in the Materials and Methods section, and some of the figures/images in the results (especially Figure 1) should be supporting these descriptions. I strongly suggest that this session be revised.  For example, in the Results section under “3.1. Characterization of the Engineered Construct”, the first lines (231-233) describes the method instead results. The same happens in other places in the session as in “3.2. Development of the VML model” it is described: To evaluate how endothelial transplantation and construct patterning influences muscle remodeling following severe muscle trauma and to outline a timeline for healing events, a VML injury was surgically created in the tibialis anterior muscle (TA, Figure 1h). Mice were divided into cohorts which received one of the following constructs: 1) Acellular Aligned, 2) EC Random, or 3) EC Aligned. Tissues were collected on day 2, 7, and 21 post-injury to histologically assess remodeling (Figure 1i).”

It happens as well in 3.3. Muscle Vascularization (lines 266-269), 3.4. Muscle Myogenesis (lines 303-307), 3.5. Muscle Collagen Content and Construct Fibrous Encapsulation (lines 378-382). It would be beneficial to retain the comprehensive account of the findings, relocating any information that does not constitute a definitive outcome to the Methods or Discussion section. In this regard, it would be advantageous to adapt and rewrite the entire Materials and Methods section, as well as the Results section. Despite that, Figures and graphics are well presented.

Discussion:

          The discussion is structured in a clear and coherent manner, with each item contributing valuable insights and reflections on the presented results. However, it would be advantageous to start with a summary of the main results obtained according to the proposed objectives.

Furthermore, it would be prudent to include a dedicated paragraph that addresses the constraints of the study.

Conclusions: The conclusion could be enhanced by a closer examination of the proposed objectives. It would be prudent to rewrite it in accordance with this advice.

References: Most of them are current and relevant to the topic.

Reviewer 2 Report

Comments and Suggestions for Authors

Current research examines the impact of endothelial cell (EC)-laden, nanofibrillar collagen constructs with aligned and random patterning on muscle regeneration following volumetric muscle loss (VML) injury in a murine model, to address the temporal changes that occur during the early stages of muscle remodeling in response to engineered therapies. The final conclusion suggests that EC-laden constructs, regardless of patterning, have limited impact on muscle repair following VML injury. On the other hand, EC-laden constructs lead to more collagen deposition and fibrotic encapsulation, indicating an immune response that could limit therapeutic efficacy. The study is relevant to the field of regenerative medicine, as effective treatments for VML are limited, and improving muscle regeneration with biomaterial-based constructs is a priority. The specific application to VML injuries and the focus on temporal changes in tissue response provides a unique angle to the research. Methodology conducted here should be rational and conclusion has been derived from solid and clear presentation. The references cited are relevant and trendy. Significance of current study has also been realized in indicating a possible repair timeline for VML injuries as well as providing future directions for subsequent studies in regenerative treatments for muscle injuries.

Round 2

Reviewer 1 Report

Comments and Suggestions for Authors

I am grateful for the opportunity to revise the article with the corrections and clarifications made, and that my initial comments have been taken into account.

After careful reading and analysis, I believe that the changes and clarifications made by the authors have considerably improved the quality and presentation of the article, with the only thing missing being the inclusion of the figure dealing with the design of the study. 

Author Response

Dear Editors of Bioengineering,

We thank the reviewer for their additional feedback regarding our manuscript (bioengineering-3319407), “Temporal Tissue Remodeling in a Volumetric Muscle Injury with Endothelial Cell-Laden Patterned Nanofibrillar Constructs.”  In response to this feedback, we have added a study design figure by moving the scanning electron microscopy data to a separate, independent figure, and rearranging the remaining sub-figures within Figure 1 accordingly.  

Please find our point-by-point response to the reviewer’s comment below. The reviewer comment is presented in bold italics, with our response following in regular text. Changes to the manuscript are highlighted in yellow.

Reviewer

I am grateful for the opportunity to revise the article with the corrections and clarifications made, and that my initial comments have been taken into account.

After careful reading and analysis, I believe that the changes and clarifications made by the authors have considerably improved the quality and presentation of the article, with the only thing missing being the inclusion of the figure dealing with the design of the study.

We thank the reviewer for their positive feedback on the improved quality and presentation of our manuscript. To further enhance clarity, we have now included a study design figure in the Methods Section by reorganizing elements of the original Figure 1 into two separate figures that are split between Methods and Results. The new Figure 1 with the study design is in the Methods section while the scanning electron microscopy data remains in the results section and is the new Figure 2. Both revised figures and their subsequent captions are included below for reference.